# Mucormycosis: A 14-Year Retrospective Study from a Tertiary Care Center in Lebanon

**DOI:** 10.3390/jof9080824

**Published:** 2023-08-03

**Authors:** Fatima Allaw, Johnny Zakhour, Sarah B. Nahhal, Karim Koussa, Elio R. Bitar, Anthony Ghanem, Martine Elbejjani, Souha S. Kanj

**Affiliations:** 1Division of Infectious Diseases, Department of Internal Medicine, American University of Beirut Medical Center, Beirut 110236, Lebanon; fballaw@gmail.com (F.A.); johnnyezakhour@gmail.com (J.Z.); sarahnahhal@gmail.com (S.B.N.); 2Faculty of Medicine, American University of Beirut, Beirut 110236, Lebanon; kjk05@mail.aub.edu (K.K.); erb07@mail.aub.edu (E.R.B.); 3Department of Otolaryngology-Head & Neck Surgery, American University of Beirut Medical Center, Beirut 110236, Lebanon; anthonyghanem01@gmail.com; 4Clinical Research Institute, Department of Internal Medicine, Faculty of Medicine, American University of Beirut, Beirut 110236, Lebanon; me158@aub.edu.lb; 5Center for Infectious Diseases Research, American University of Beirut, Beirut 110236, Lebanon

**Keywords:** mucormycosis, hematological malignancy, diabetes mellitus, rhino-orbital-cerebral mucormycosis, amphotericin B, COVID-19, Lebanon

## Abstract

Mucormycosis (MCM) is a serious invasive fungal disease (IFD) that is associated with high mortality, particularly in immunocompromised patients. A global surge in MCM cases was reported with the COVID-19 pandemic. We analyzed all recorded cases of MCM at the American University of Beirut Medical Center (AUBMC), a tertiary care center in Lebanon, over 14 years. We aimed to identify the incidence, seasonal variation, clinical characteristics of the patients, and predictors of mortality. We conducted a retrospective chart review between 1 January 2008 and 1 January 2023. All patients with proven or probable MCM were included in the study. Proven or probable MCM was defined by positive histopathology and/or positive cultures. A total of 43 patients were identified as having MCM. Their median age was 53 years, and the majority were males (58.1%). Most of the cases were diagnosed in the autumn season. In total, 67.4% of the patients had hematological malignancies (HMs), and 34.9% had uncontrolled diabetes mellitus (DM). The most common site of involvement was rhino-orbital-cerebral MCM (ROCM) (74%). The annual cases of MCM per 100,000 patient days increased markedly during the years of the COVID-19 pandemic (from 0 to 4.4 cases/100,000 patient days to 7.5 cases/100,000 during 2020 and 2021). Liposomal amphotericin (Ampho) B was used as a first-line agent in most of the patients (86%). The median duration of total in-hospital antifungal therapy was 21 days and 51.2% of the patients received step-down therapy with azoles. Surgical debridement and isolated ROCM were significantly associated with survival (*p*-value: 0.02 and <0.001, respectively). All-cause mortality was 46.7%, with chronic renal disease being significantly associated with mortality (*p*-value < 0.05). The incidence of MCM has been increasing at our institution, particularly since the COVID-19 pandemic. Early diagnosis, treatment, and surgical debridement improve patient outcomes and overall survival.

## 1. Introduction

Mucormycosis (MCM) is a life-threatening invasive fungal disease (IFD) caused by fungi of the order Mucorales [1,2]. The most frequently reported pathogens include *Rhizopus* spp., *Mucor* spp., and *Lichtheinmia*, followed by *Rhizomucor* spp., *Cunninghamella* spp., *Apophysomyces* spp., and *Saksenaea* spp. [3,4,5]. Various infection rates have been reported for different institutions and for high-risk populations [2].

Infection usually occurs through the inhalation of fungal spores into the respiratory tract, ingestion through the gastrointestinal tract, or through direct inoculation to areas of trauma [6]. MCM occurs mostly in patients with an impaired immune status, including uncontrolled diabetes mellitus (DM), transplant recipients, patients with hematologic malignancies (HMs), and neutropenia. However, immunocompetent hosts can also develop an infection through the direct inoculation of organisms [2]. More recently, MCM has been increasingly reported in COVID-19-infected patients, particularly following the receipt of steroids or other immunosuppressant agents such as tocilizumab [7].

The diagnosis of MCM can be challenging as it relies on tissue sampling with specific findings on histopathologic examination and culture with no available antigen detection tests as is the case for other deep-seated fungi [8]. The hallmark of MCM is angioinvasion, resulting in rapid progression of the infection and extensive tissue necrosis; often leading to the invasion of adjacent organs and blood vessels [9,10,11,12]. The spectrum of infection ranges from rhino-orbito-cerebral MCM (ROCM) to pulmonary, cutaneous, gastrointestinal, and disseminated MCM [12].

During the COVID-19 pandemic, a global surge of MCM cases was reported with the majority of cases stemming from India [13]. Numerous cases in low- and middle-income countries (LMIC) may have gone unreported to healthcare facilities, resulting in limited representation in the published literature. Moreover, there is limited knowledge regarding the prevalence of MCM in the Arab countries [14]. The largest study to assess the incidence of MCM in Lebanon was conducted at the American University of Beirut Medical Center (AUBMC), a tertiary referral center, where a total of 20 cases of MCM were identified over a 10-year period from 2008 to 2017. The reported number of MCM cases per 10,000 hospital admissions was 1.18 in 2017, increasing from 0.47 in 2008 [15].

Since then, and especially after the COVID-19 pandemic, we have noticed an increase in the number of cases of MCM. To better understand this observation, we conducted a retrospective study at AUBMC between 1 January 2008 and 1 January 2023. We herein describe the incidence of MCM, seasonal variations, clinical characteristics of infected patients, and the predictors of mortality.

## 2. Research Design and Methods

### 2.1. Study Design and Setting

We conducted a retrospective study that included all patients that were admitted to the AUBMC between 1 January 2008 and 1 January 2023 and were diagnosed with MCM.

AUBMC is an academic tertiary care center with a capacity of 420 beds. It is a national and regional referral center in Beirut, Lebanon. This institution provides inpatient medical care to over 25,000 patients a year. It provides specialized medical and surgical services including a specialized hematology and oncology service and a busy bone marrow transplant service performing over 100 transplants per year.

### 2.2. Population and Data Collection

Using the medical records department at AUBMC, we identified all cases of MCM that were diagnosed between 1 January 2008 and 1 January 2023. Since 2018, AUBMC has moved to an electronic medical records system.

The collected data included: demographics (age, sex), comorbidities (DM, chronic renal disease), immunocompromising conditions (solid organ malignancy, HM, chemotherapy, or immunotherapy within the previous 30 days, hematopoietic stem cell transplant (HSCT) or solid organ transplantation (SOT), neutropenia), COVID-19 infection and treatment with immunosuppressant agents (steroids, tocilizumab, baricitinib). Imaging findings of Computed Tomography (CT) scans were also collected as well as microbiological results and pathology reports. We also collected data regarding the management of patients including surgical debridement from operative reports, antifungal agents used with their respective durations, and any documented adverse events.

We also aimed to identify the annual and monthly incidence of MCM per 100,000 patient days between the years 2008 and 2023. Data on annual admissions to all medical and surgical wards (including pediatrics) to AUBMC was obtained to calculate the annual incidence per 100,000 patient days.

We also compared patients who survived and those who were deceased within the first 30 days of admission to identify predictors of mortality.

### 2.3. Microbiological Definitions

We included both proven and probable cases of MCM. The patients were classified as having probable or proven disease based on the global guideline for the diagnosis and management of MCM of the European Confederation of Medical Mycology in cooperation with the Mycoses Study Group Education and Research Consortium [2]. A case is classified as “proven MCM” when there is definite microbiological evidence confirmed by culture from a sterile site while “probable MCM” requires a positive culture from a non-sterile site combined with clinical and radiological evidence of the disease. Histopathology was consistent with MCM when broad, non-septate hyphae with irregular right-angle branching were visualized on microscopy. Cultures and speciation were not always available. For pulmonary MCM, pathology and culture were performed on samples from endobronchial aspirates, broncho-alveolar lavages (BAL), or lung biopsy. Molecular methods for fungi identification are not available at our medical center.

All cases of COVID-19 were confirmed using a real-time reverse transcription polymerase chain reaction (RT-PCR) performed on nasal swabs, tracheal aspirates, BAL, or other respiratory samples.

### 2.4. Statistical Analysis

Data management and analysis were conducted using IBM SPSS version 28 (IBM, New York, NY, USA). Categorical data were presented using count (percent), while continuous data were presented using medians and interquartile ranges (IQR). Associations between categorical variables and the outcome variable (30-day in-hospital mortality) were assessed using the Chi-square test or Fisher’s exact test when more than 20% of expected cell counts were below 5. Associations between continuous variables and the outcome variables (30-day in-hospital mortality) were assessed using the Mann–Whitney test. Significance was two-tailed and was set at α < 0.05. A Kaplan–Meier estimate using the log-rank test was used to analyze survival.

### 2.5. Ethical Considerations

The study was approved by the institutional review board (IRB) at AUBMC (Protocol number: BIO-2022-0364). Since this was a retrospective chart review, patients’ consent was waived. The study posed no risk to patients and its potential benefits outweighed the potential risks.

## 3. Results

### 3.1. Baseline Characteristics of the Study Population

We identified 43 cases of patients with MCM admitted to AUBMC between January 2008 and January 2023. The median age of our population was 53 and 58.1% of them were males and had a median Charlson Comorbidity Index of 2. Two-thirds of our population (67.4%) had an HM. One-third (34.9%) had DM. We identified 5 cases of COVID-19-associated MCM (CAM).

The most common presenting signs and symptoms were fever (55.8%) and symptoms and signs of sinusitis (55.8%). A necrotic eschar was seen in 39.5% of cases. ROCM was the most common presentation and accounted for 32 cases (74.4%). Of those, 84.4% had isolated ROCM, while the remaining cases had ROCM associated with either pulmonary or skin and soft tissue (SSTI) MCM. Common imaging findings for ROCM included soft tissue edema (40.6%) and mucoperiosteal thickening of the sinuses (40.6%). Imaging findings of pulmonary MCM consisted mostly of consolidation (66.6%) and infiltrates (50.0%). Interestingly, the reverse halo sign was not seen in any of the cases. A biopsy was performed in 39 patients of which 82.1% had, on microscopy, non-septate hyphae branching at 90 degrees. The most commonly biopsied sites were sino-nasal (56.4%), soft tissue (20.5%), and palatal (12.8%). On the other hand, culture was positive in only 10 of those cases.

Regarding the management, surgical debridement was performed in 72.1% of all MCM cases, mostly in the ROCM and the SSTI MCM cases, with a median time from diagnosis to surgery of 1 day. Most of the patients received initial treatment with a lipid formulation of amphotericin (Ampho) B, mostly liposomal Ampho B, for a median duration of 21 days. A total of 14% of the patients developed nephrotoxicity, and 23.3% had electrolytes abnormalities while on antifungal treatment (Appendix A). Among those who underwent step-down antifungal therapy, posaconazole was the most commonly prescribed agent (72.4%). Thirty-day in-hospital mortality was 46.5% with a median time from diagnosis to death of 20.5 days. In those who survived, the median time from diagnosis to discharge was 29 days (IQR = 24.0–42.0 days) (Appendix A).

### 3.2. Seasonal and Annual Variations in the Incidence of Mucormycosis

We calculated the annual incidence of MCM per 100,000 patient days. While the incidence fluctuated between 0 and 4.4 cases/100,000 patient days, it reached 7.5 cases/100,000 patient days during the years 2020 and 2021 following the COVID-19 pandemic (Figure 1).

The highest number of cases of MCM was recorded during the month of October. Over half of the cases occurred during the period extending from the months of August till November corresponding to the late summer and autumn season (Figure 2).

### 3.3. Predictors of Mortality from Mucormycosis

When comparing patients who survived to those who were deceased, we found that isolated ROCM and surgical debridement were significantly associated with survival on bivariate analysis. In addition, higher survival rates were seen in male patients and in patients with recent antifungal use in the past 30 days prior to diagnosis; however, this was not statistically significant. On the other hand, chronic renal disease, as a comorbidity present on admission, was significantly associated with mortality (Table 1).

Although female sex, neutropenia, SOT, ROCM and lung MCM, SSTI MCM, and recent corticosteroid treatment in the past 30 days prior to the diagnosis of MCM were associated with higher mortality, this was not statistically significant. Furthermore, compared to those who survived (2), a higher median Charlson Comorbidity Index was seen in patients who were deceased (4) (*p*-value: 0.076).

In-hospital 30-day mortality was significantly reduced when surgical debridement of the site of MCM was performed, mostly in cases of ROCM and SSTI MCM (Figure 3).

## 4. Discussion

Our study provides informative insights into the trends and clinical characteristics of MCM in patients admitted to a tertiary care center in Lebanon over a 14-year period. Despite the limited number of infected patients, several key aspects of MCM could be highlighted.

A cluster of MCM cases was observed in 2020 and 2021 correlating with the COVID-19 pandemic as has been described in other case series [16,17]. Moreover, it is worth mentioning that fungi of the order Mucorales are recognized as significant contributors to wound infections in various settings including natural disasters, and other forms of civilian trauma [18,19,20]. In fact, the Beirut Port Blast happened in August 2020 and might have contributed to a rise in the environmental fungal load of Mucorales spores possibly leading to an increase in the number of cases admitted to AUBMC, which is located in the city of Beirut. In addition, our study shows a low culture yield compared to other published studies [4,21,22,23]. The discordance between histopathological diagnosis and fungal culture results has been well described in the literature [23]. Grinding or homogenizing tissue samples could have potentially damaged the fragile hyphae, leading to negative culture outcomes. Thus, it is crucial for clinicians and microbiologists to closely collaborate with each other to ensure proper handling of the specimen [24,25].

Most of the reported cases in this study were seen in the late summer and autumn. Previous studies from Lebanon also showed this seasonal predominance of cases [15,26]. This seasonal trend is likely related to the role of the predisposing environmental factors. In a study from Iran, positive soil samples of Mucorales were most frequently detected in autumn (43.2%) [27]. The warm weather in Lebanon, coupled with increased humidity extending to the autumn season, may be associated with an increase in Mucorales spores’ concentration [28,29].

The most known risk factors for developing MCM are DM, HMs, other types of cancer, organ transplantation, long-term neutropenia, the use of corticosteroids, traumatic injuries, and excess iron levels [23,30]. In Iran and India, DM has been consistently identified as the most frequently reported risk factor in patients with MCM, while in Europe and USA, HM has emerged as the most common underlying disease [4,31,32,33,34,35]. A recent 11-year multi-center study conducted in Saudi Arabia showed comparable proportions of DM and HMs among the patient population diagnosed with MCM (48% and 42.4%, respectively) [36]. In our study, almost two-thirds of our patients had an HM and one-third had DM. The prevalence of DM has been increasing at a faster rate in LMIC compared to high-income countries, which might explain the consequent expected increase in the incidence of MCM in these countries [37]. Additionally, poorly controlled DM is more likely in LMIC and contributes to the differences seen in the incidence of MCM [38]. Although Lebanon is considered a LMIC, the elevated incidence of MCM in those with an HM could potentially be attributed to AUBMC’s position as a referral center for complex cases of HMs and HSCTs, as previously described in a case series from the same tertiary care center [15].

CAM has recently gained attention as a rare but severe complication in COVID-19 patients, likely associated with the widespread use of corticosteroids, uncontrolled hyperglycemia, broad-spectrum antibiotics, and interleukin 6 inhibitors [7,21,39,40,41,42]. A cumulative dose greater than 600 mg of prednisone and 2 to 7 g of methylprednisolone has been associated with an increased risk of MCM in immunocompromised patients [43]. The majority of reported cases of CAM worldwide have originated from the Indian population [44]. Data regarding CAM in the Arab region primarily stems from Egypt where several reports have described cases of COVID-19 associated with ROCM, as well as the high prevalence of DM and the use of corticosteroids in these patients [44,45,46,47]. In addition, one case series from a secondary hospital in Oman reported 10 cases of COVID-19 and ROCM, all of which were observed in patients with poorly controlled DM [48]. It has been shown that the median duration for patients with COVID-19 to develop MCM is around 10–15 days from COVID-19 diagnosis [39,49]; however, delayed MCM has also been reported [7]. Although our study included only 5 cases, this small number could be attributed to the limited sample size. All the 5 patients had received high dose corticosteroids and only 2 of them had type II DM with uncontrolled blood sugar levels. Multiple conjectures exist regarding the causative factors of MCM in COVID-19. First, the administration of corticosteroids leads to elevated blood sugar levels while diminishing the phagocytic cells, thereby heightening the susceptibility to MCM [50]. In addition, it has been suggested that the SARS-CoV-2 virus can impair pancreatic islet cells, potentially inducing new onset DM or exacerbating hyperglycemia in patients with pre-existing DM [39]. In India, it was hypothesized that the hospitals’ air quality could have potentially played a role in some nosocomial outbreaks of CAM [21]. It should be mentioned that routine air sampling has been performed at our facility in the oncology and the bone marrow transplant units over the past 20 years with the occasional growth of *Aspergillus* spp. but never of Mucorales (data from the infection control program at AUBMC). A recently published meta-analysis suggests that prior pulmonary infection with *Aspergillus* spp. and tocilizumab use in patients with COVID-19 are risk factors for MCM [51].

Associations between the site of MCM infection and various underlying medical conditions have been described in the literature. For example, HM is frequently linked to pulmonary involvement, whereas in diabetic patients, sinus disease is more commonly seen than other sites of involvement [4,52]. Globally, including the Middle East region, ROCM is the most frequently reported site of infection [53,54]. Similarly, in this study, the most commonly involved site was ROCM with symptoms and signs of sinusitis being the most commonly presenting clinical picture. In most of the patients, the diagnosis was suspected based on the clinical picture and the CT scan findings. The symptoms associated with ROCM are often nonspecific and can vary in terms of range and severity. The nasal or palatine manifestations of MCM include the presence of grey or reddish mucosa, which subsequently progresses to the formation of black areas of eschar as necrosis develops. Eschar can be observed in the nasal septum, palate, eyelid, face, or orbital regions. However, the absence of eschar does not necessarily exclude the possibility of ROCM [30,54,55]. A CT scan offers certain advantages over magnetic resonance imaging (MRI) as it can detect bony erosion and necrosis, although such findings typically indicate an advanced stage of the disease with a poor prognosis. A contrast CT scan can also reveal evidence of cavernous sinus thrombosis and vascular enhancement. On the other hand, MRI is particularly useful in the early detection of cranial vascular invasion, making it more beneficial than CT brain scans in these aspects [56]. In our study, a CT scan was the preferred modality over MRI. The most frequently observed findings of ROCM were orbital invasion, followed by mucoperiosteal thickening and severe soft tissue edema. Approximately half of the diagnosed patients with ROCM exhibited orbital involvement, which aligns with findings from the literature [57,58,59]. The pterygopalatine fossa serves as a crucial pathway for the spread of MCM to the orbital, facial, and intracranial regions [60].

Pulmonary MCM mainly occurs in immunocompromised patients with the main risk factor being an HM [30]. Symptoms are not specific and can range from persistent fever to cough and hemoptysis [61]. Moreover, after diagnosing pulmonary MCM, clinicians should actively look for additional invasive signs such as ROCM, or skin or gastrointestinal symptoms [62]. In our study, 3 of the patients with pulmonary MCM also had concomitant ROCM on diagnosis. The presence of the reverse halo sign, which is characterized by a concentrated ground-glass opacity (GGO) surrounded by a ring or crescent-shaped consolidation, is typically considered a strong indicator of pulmonary MCM [63]. However, it is important to note that this sign is often an early manifestation and may not be present in all patients [64]. For instance, in a recent study conducted in France, it was found that condensations and GGO were the most common types of lesions observed (58% and 65%, respectively), which is similar to our findings (50% and 66.6%, respectively) [65].

SSTI MCM, ranking as the third most prevalent form of MCM, after ROCM and pulmonary MCM, was examined in a recent systematic review by Skiada et al. The study revealed that DM was the leading underlying disease in SSTI MCM cases, accounting for 20% of the reported cases, followed by HMs at 15.7%. Notably, a significant finding was that 39.6% of the documented cases involved immunocompetent patients [66]. In our study, the 5 cases of isolated SSTI MCM were identified in patients with an HM, while only 2 patients had DM. One of our patients developed MCM on the anterior shins of the legs following prolonged exposure to the sequential compression device (SCD) used for deep vein thrombosis prevention. Whether MCM was caused by a direct trauma from the SCD remains uncertain. Interestingly, two studies from Saudi Arabia showed that the most common site of infection was cutaneous MCM [36,67]. Elzein et al. showed a notable occurrence of cutaneous MCM associated with significant trauma, particularly from motor vehicle collisions (38.9%) [67].

When examining factors correlating with outcomes in our series, it was found that ROCM and surgical debridement were significantly associated with better survival. It is likely that patients with ROCM were diagnosed at an early stage, allowing for prompt surgical debridement and adequate treatment, which contributed to higher rates of survival, as opposed to patients with MCM in other locations. For instance, the median time from diagnosis till surgery was 1 day in our study. The morbidity and mortality in ROCM are primarily influenced by the timing of initiating intravenous Ampho B therapy, and the timing of surgical debridement, and any delay in implementing these measures, can have an impact on survival outcomes [68,69,70]. Additional factors associated with poor survival from the literature were bilateral sinus involvement on initial diagnosis, brain and cavernous sinus involvement, and renal disorder [56]. All patients received formulations of Ampho B, mostly liposomal Ampho B, at a dose varying between 5 and 10 mg/kg/day except two who had a poor prognosis on diagnosis and opted for palliative care with no antifungal treatment. In resource-constrained settings, either Ampho B deoxycholate or Ampho B lipid complex (Abelcet) can also be used. In total, 86% of our patients received liposomal Ampho B (Ambisome) for a mean duration of 21 days. Around half of them underwent step-down therapy to posaconazole after a median of 21 days. Prolonged step-down therapy is warranted for months [2]. Chronic renal disease upon diagnosis was significantly associated with mortality in our study. On the other hand, a recent meta-analysis showed that the mortality in patients with initial chronic renal disease has decreased from 52% to 19% over 20 years, likely due to the administration of liposomal Ampho B, which is less nephrotoxic than Ampho B [69].

The strength of our study is the wide clinical data and characteristics presented on patients with MCM from Lebanon over a 14-year period including pre- and post-COVID-19. However, several limitations are present. First, the retrospective nature of the study introduces the possibility of bias during data collection. Second, the small number of included patients may have influenced the generalizability of the conclusions. Third, the fact that our tertiary care center serves as a referral facility for HMs and HSCT treatment could have potentially impacted the findings, thereby not being a true reflection of the epidemiology of MCM in Lebanon.

## 5. Conclusions

The incidence of MCM has been increasing over the past 14 years at our institution with a predominance in the autumn months. The epidemiology and clinical characteristics of MCM have undergone changes in Lebanon and globally due to the impact of the recent COVID-19 pandemic and the increase in chronic noncommunicable diseases. Managing this disease poses significant challenges given its high morbidity and mortality rates, necessitating a multidisciplinary approach for evaluation and treatment. Understanding the specific epidemiological patterns in each country, particularly in LMIC where resources are limited, is essential as it guides decisions regarding antifungal prophylaxis, early detection, and prompt management with surgical debridement and empirical antifungal therapy.

## Figures and Tables

**Figure 1 jof-09-00824-f001:**
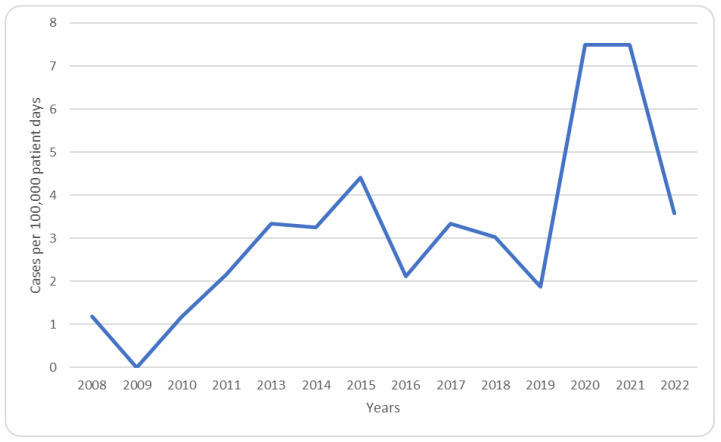
Annual cases of mucormycosis per 100,000 patient days.

**Figure 2 jof-09-00824-f002:**
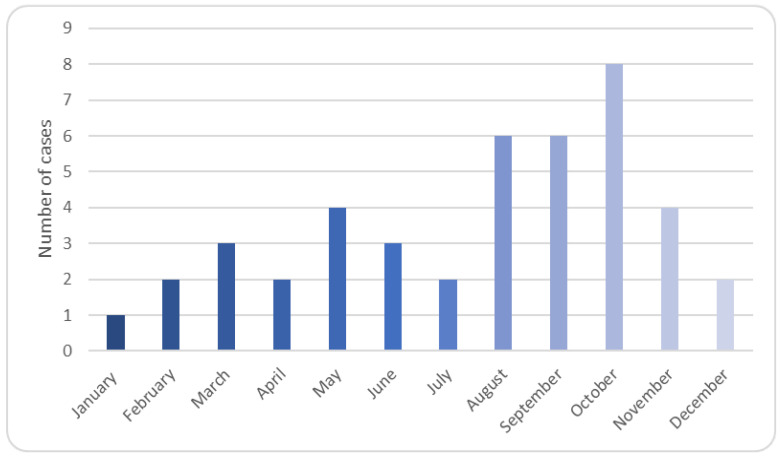
Monthly distribution of mucormycosis cases.

**Figure 3 jof-09-00824-f003:**
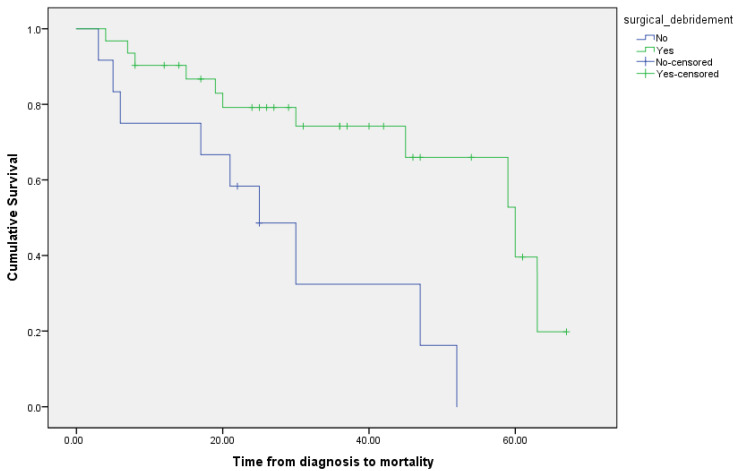
Kaplan–Meier survival plot for time from diagnosis to mortality according to surgery status.

**Table 1 jof-09-00824-t001:** Predictors of mortality in patients with mucormycosis. BMT, bone marrow transplant; GVHD, graft-versus-host disease; IQR, interquartile range; MCM, mucormycosis; qSOFA, quick Sequential Organ Failure Assessment; ROCM, rhino-orbito-cerebral mucormycosis; SSTI, skin and soft tissue infection.

	Dead (N = 20)	Survived (N = 23)	*p*-Value
Age (median, IQR)	56 (35–65)	50 (38–58)	0.592
Male Female	9 (45.0%) 11 (55.0%)	16 (69.6%) 7 (30.4%)	0.103
Charlson Comorbidity index	4 (2–5)	2 (2–4)	0.076
Diabetes mellitus	8 (40.0%)	7 (30.4%)	0.512
Chronic renal disease	5 (25.0%)	0 (0.0%)	0.016
Immunocompromised (solid organ malignancy, hematological malignancy, chemotherapy, or immunotherapy within the previous 30 days, hematopoietic stem cell transplant or solid organ transplantation, and neutropenia)	20 (100.0%)	18 (78.3%)	0.051
Hematologic malignancy	13 (65.0%)	16 (69.6%)	0.750
Solid organ malignancy	3 (15.0%)	0 (0%)	0.092
Transplant	4 (20.0%)	2 (8.7%)	0.393
Neutropenia	12 (60.0%)	9 (39.1%)	0.172
Solid organ transplant	0 (0.0%)	1 (4.3%)	1
BMT	5 (25.0%)	4 (17.4%)	0.711
GVHD	3 (15.0%)	2 (8.7%)	0.650
Recent corticosteroids in the past 30 days prior to diagnosis	13 (65.0%)	9 (39.1%)	0.091
Recent chemotherapy in the past 30 days prior to diagnosis	15 (75.0%)	15 (65.2%)	0.486
Recent immunotherapy in the past 30 days prior to diagnosis	1 (5.0%)	1 (4.3%)	1
Auto-immune disease	4 (20.0%)	1 (4.3%)	0.110
COVID-19	3 (15.0%)	2 (8.7%)	0.650
Recent antifungals in the past 30 days prior to diagnosis	7 (35.0%)	12 (52.2%)	0.258
Voriconazole prophylaxis	7 (35.0%)	9 (39.1%)	0.780
qSOFA (median, IQR)	0 (0–2)	0 (0–0)	0.309
Isolated ROCM	7 (35.0%)	20 (87.0%)	<0.001
Isolated pulmonary mucormycosis	1 (5.0%)	2 (8.7%)	0.554
ROCM and lung mucormycosis	3 (15.0%)	0 (0.0%)	0.092
ROCM and SSTI MCM	2 (10.0%)	0 (0.0%)	0.210
SSTI MCM	5 (25.0%)	1 (4.3%)	0.065
Surgical debridement for MCM	11 (55.0%)	20 (27.0%)	0.020
Time from diagnosis to surgery in days (median, IQR)	0 (0–4)	1 (0–2)	0.902

## Data Availability

The datasets generated during and/or analyzed during the current study are not publicly available but are available from the corresponding author upon reasonable request.

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
