# Peer review of "Mucormycosis: A 14-Year Retrospective Study from a Tertiary Care Center in Lebanon"

_jof, 2023, doi:10.3390/jof9080824_

Round 1
Reviewer 1 Report
The overall manuscript is well written.
there are some edits is recommended and comments as follow.
- Line 23 correct the date to be 2023 not 2022
- From line 81--86; what is the purpose of adding this information in the manuscript? I don’t see any direct or indirect relation to the study. my recommendation is to remove unnecessary information about the medical center.
- line 100-101; my recommendation is to remove this sentence. There is a redundance, you don’t need to mention the aim as its pinpointed in line 101-103.
- Section 2.3; it’s recommended to be rephrased for better understanding and clearly show the difference between proven and probable case of MCM.
- line 171- 173; it’s not clear if you mean the cases from August until November are contributing in 1/2 of cases or only the two months (August, and November). depending on your answer you can decide if you remove the word "respectively" or not.
- In table 1; the * is not needed in Immunocompromised category, also add the type of data presentation if its Median or mean. The abbreviation of qSOFA needs to be added. Finally, please add the data for female sex in the table.
- In section 3.3 of results (line 178- 191): you mentioned what is negatively correlated or positively corelated to the survival that is statistically significant. and you added the negatively regulated but not significant. to complete the pattern just add the positively correlated to survival but not significant (antifungal, male sex).
- Line 186 add the rest of categories that were associated with higher mortality but not significant (Solid organ malignancy, ROCM and lung Mucormycosis, and SSTI MCM).
- Line 199; you can substitute the word valuable with informative or other lower toned word.
- if there any data regarding identification of species of Mucorales in these 53 cases that will be a good add and give some information about the species prevalence in Lebanon.
Reviewer 2 Report
The manuscript is interesting and very well presented. I have only one question: Do you have more data about the isolated fungi? Only the genera are presented. Were any molecular methods used?
